# Amphibians and Reptiles of the Montagne des Français: An Update of the Distribution and Regional Endemicity

**DOI:** 10.3390/ani13213361

**Published:** 2023-10-29

**Authors:** Herizo Oninjatovo Radonirina, Bernard Randriamahatantsoa, Nirhy H. C. Rabibisoa, Christopher John Raxworthy

**Affiliations:** 1Doctoral School of Natural Ecosystems, University of Mahajanga, Mahajanga 401, Madagascar; 2Sciences de la Vie et de l’Environnement, Faculté des Sciences, de Technologies et de l’Environnement, University of Mahajanga, BP 652, Mahajanga 401, Madagascar; bernardzoo01@gmail.com; 3American Museum of Natural History, Department of Herpetology, New York, NY 10024, USA; rax@amnh.org

**Keywords:** amphibians, reptiles, Montagne des Français, update, distribution, regional endemicity

## Abstract

**Simple Summary:**

In 2014 and 2020, we conducted research in the Montagne des Français Protected Area. This area is recognized for its remarkable fauna and high rate of endemism. The objective of this study was to prioritize areas and determine the diversity and distribution of herpetofauna species, as well as their conservation status, endemism, and threats. In this study, we compared species richness between two surveys conducted in 2014 and 2020. We found that three species of amphibians and one reptile species had a new distribution. The 2020 survey also discovered *Langaha pseudoalluaudi*, a snake that has not been observed since 2007. The information gathered from this survey could be of use to site managers in the revision of conservation management plans.

**Abstract:**

The harmonious landscape of Montagne des Français is a protected area in the far north of Madagascar. Our herpetofauna surveys were conducted on the eastern and western slopes according to habitat variations within the massif for 2014 and 2020. Our research updates the herpetofauna species richness, spatial distribution ranges, and ecological guilds within the protected area. We used direct opportunistic observations, systematic refuge examinations, and pitfall traps with drift fences at three sites to sample animals. Nineteen amphibian and fifty reptile species were recorded during this study. Three amphibian species and one reptile species, in addition to the snake *Langaha pseudoalluaudi*, last recorded in 2007, were discovered at Montagne des Français. Here, we present a database update for the herpetofauna species from Montagne des Français and provide a specific morphological description of the morphospecies considered as a new extension or localized distribution. This new database can help site managers develop new strategic conservation plans in response to habitat modification.

## 1. Introduction

Madagascar is recognized globally as a biodiversity hotspot [1]. High levels of endemism and diversity are observed due to its isolation [2,3,4]. The creation of protected areas is underway to conserve biodiversity and ecosystem services [5]. The island also has a highly endemic herpetofauna, with 90% of the species found nowhere else [6]. Montagne des Français, which is one of Madagascar’s protected areas, is situated in the northern region. The site is a limestone massif [7] and is home to an exceptional community of herpetofauna [8], including several endemic species [9].

Unfortunately, the biodiversity of the flora and fauna in this protected area has been affected by human activities. These anthropogenic pressures have caused habitat loss and significant alterations to the ecosystem [7,10], resulting in landscape erosion. Currently, dry forests are one of the most threatened ecosystems on the island as they are subjected to frequent fires [11,12]. Among vertebrates, herpetofauna are especially sensitive to habitat alterations and disturbances [13,14]. In addition, numerous studies have shown that species within the herpetofauna category are among the most endangered vertebrates on Earth [15,16,17]. This is typically observed in tropical regions [18]. Although research efforts have been ongoing for decades, the inventory of Madagascar’s herpetofauna remains incomplete [19,20].

Research in Montagne des Français has provided incomplete data on the distribution and status of species [7,20,21]; however, ongoing field studies continue to collect new information. A large body of literature indicates that Montagne des Français is home to many reptile and amphibian species. Among them are *Heteroliodon fohy* [22] and *Thamnosophis martae* [23] snakes, along with *Brookesia tristis* [24]. The gecko *Paroedura lohatsara* is also present in this locality, along with various frog species, including *Stumpffia staffordi* [22,25] and *Mantella viridis* [26,27]. The majority of these species are microendemic or regionally endemic, meaning that they are exceptionally or primarily found in the Montagne des Français and its neighboring regions.

Many of the species under consideration have recently undergone assessments revealing that they are “Critically Endangered” or “Endangered”, highlighting their precarious position regarding extinction. Human activities, such as agriculture and logging, have severely reduced their habitats and caused fragmentation and degradation, all of which pose significant threats to their continued survival. The distribution maps of several species suggest that a biogeographical border exists between Montagne des Français/Orangea and limestone massifs situated to the south, underscoring the importance of safeguarding and managing Montagne des Français as a critical habitat for these reptiles [28]. Furthermore, species inventories have only been conducted in a few areas of Montagne des Français. In this study, we investigated two sites that are much less visited, Ampitiliantsambo and Sahabedara.

This article presents recent findings on the biodiversity, biogeography, and endemicity of herpetofauna residing in Montagne des Français. Field surveys were conducted on both the eastern and western slopes of the mountain in 2014 and 2020, incorporating ecological parameters, such as habitat type, stream flow, and degree of degradation. The database on biodiversity resulting from the two studies of amphibians and reptiles serves not only to prioritize species but also to ease pressure on the protected core area and aid the site manager in executing the new management plan. These new data can also contribute to the implementation of a new conservation strategy, given the many threats that surround this protected area [29,30]. Optimal management of protected areas requires access to scientific data and comprehensive knowledge of biodiversity, which should be incorporated into the design of conservation and management strategies [31]. Biological inventories are highly effective means of acquiring such information [32].

## 2. Materials and Methods

### 2.1. Study Site 

Montagne des Français is a limestone plateau intersected by narrow canyons, located in northern Madagascar, between 12°18 and 12°27′ latitudes south and 49°21 and 49°23′ longitudes east, with a maximum elevation of 425 m asl. The forest massif is very close to the city of Antsiranana, measuring 6049 ha (Figure 1). This site has had a fully protected status as a Category V new protected area since 28 April 2015 (Decree N°. 2015-780) and is managed by the NGO “Service d’Appui pour la Gestion de l’Environment” which is located in the city of Antsiranana.

Sahabedara is located at an altitude of 159 m between 049°21′41.7″ E and 12°23′05.8″ S. The area consists of a gallery forest with sandy and rocky terrains. The forest is moderately degraded, with an open canopy, and is situated outside the core area of Montagne des Français.Ampitiliantsambo lies between 049°23′05.5″ E and 12°23′16.6″ S at an altitude of 204 m. It supports a natural semi-deciduous forest and an open-canopy fragmented forest. This site is situated in the peripheral area of Montagne des Français.Andavakoera is located between 049°20′58.1″ E and 12°19′49.6″ S at an altitude of 173 m. It consists of a Sambirano forest or a relict gallery forest with sand and rocks. The site represents the core area of Montagne des Français PA.

**Figure 1 animals-13-03361-f001:**
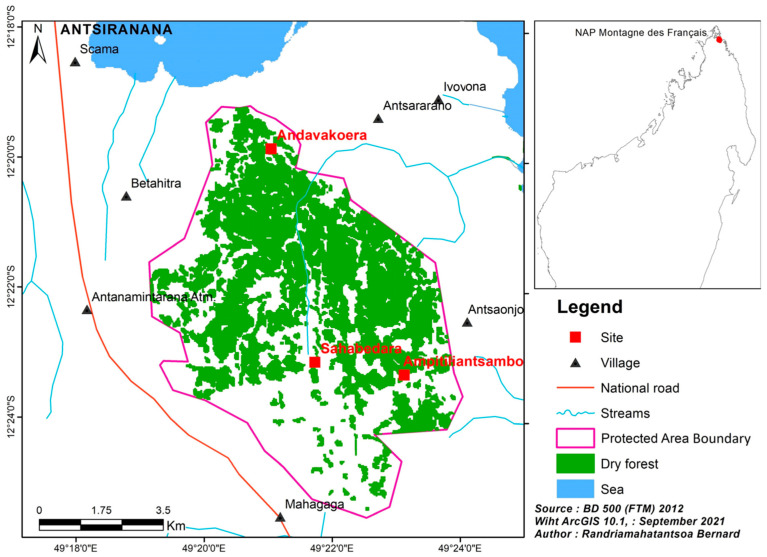
Map of the study sites in Montagne des Français.

### 2.2. Data Collection

We conducted a biological inventory at the Andavakoera site from January 13 to 24 in 2014, and from May 3 to 24 in 2020. The first survey focused only on the Andavakoera site and the second on Sahabedara, Ampitiliantsambo, and Andavakoera (Figure 1). The Andavakoera site is situated in the core area, while Sahabedara and Ampitiliantsambo are in the peripheral areas. In 2014, a group of four individuals conducted ten days of fieldwork, whereas in 2020, a group of seven individuals conducted six days of fieldwork per site. We established four transects to ensure that the different habitat types at each site were surveyed. The transect length ranged from 1500 m to 3000 m. We determined each transect based on the type of environment, plant and geological formations, topography (valley bottoms, flanks, canyons, and summits), and degree of degradation. The field guide developed by [8] was used to identify species.

During these expeditions, three standard field techniques were used to sample the herpetofauna community: 

(1) Pitfall trapping: We used pitfall traps made up of buckets (275 mm deep, 290 mm top internal diameter, and 220 mm bottom internal diameter). To allow water to drain, we removed the handles and made a few tiny holes at the bottom. We buried the buckets with their upper rims flushed with the earth at intervals of 10 m along a drift fence. We used 100 m long by 0.5 m high black plastic sheeting to create the drift fence. We positioned the fence to cut through the center of each pitfall trap and fastened it vertically to short wooden pegs. To prevent individuals from passing beneath the barrier, we dug the bottom of the plastic approximately 50 mm deep into the ground. Each day, in the early morning and late afternoon, we checked the pitfall traps and removed all the captives. For eight nights, we employed three drift fence pitfall arrays at each location.

(2) Refuge examinations: We conducted active searches both during the day and night in all possible habitats and altitudes. The majority of the searches, which lasted between 30 min and three hours, were concentrated in the interior of the forest near pathways and trails for cryptic reptile species and arboreal frogs, as well as in streams and the related riparian vegetation for amphibians. Examining refuges are suitable for both active daytime and nocturnal resting animals.

(3) Opportunistic searching [33]: This method complemented the refuge examinations and was usually carried out at the same time and along the same transect. This involved creating an inventory of amphibians and reptiles perceived along the transect. The transect was visited once per day to mitigate disturbance to the animals. As certain groups of species are nocturnal, observations were also conducted at night.

### 2.3. Data Analysis

Population structure was determined by examining several parameters, including the number of species present at each site, their abundance, status, ecological distribution, and habitat frequency. Concerning the habitat description, we used two forest types according to habitat loss, which was also used in [34]. The first one represents a forest with a weak pressure (noted F) where there are more big trees (≥25 m), and the second one has numerous observed anthropogenic activities as well as fewer big trees (noted A). We used a classification system similar to that developed by [35] to categorize species based on their abundance. This system is as follows: abundant (encountered regularly in large numbers throughout the year), common (encountered regularly year-round), infrequent (encountered unpredictably and in small numbers), and rare (rarely encountered). All identifications of the species are only based on morphological analysis. For frogs, we used stable morphological features like mensuration, iris peripheral coloration, and webbing formula; and for Squamata, we used mensuration, characteristics of rostral appendix, and scale and spine features.

## 3. Results

### 3.1. Species Richness 

We recorded a total of 19 amphibian and 50 reptile species at Montagne des Français during the two surveys of 2014 and 2020, which provided an overall herpetofaunal diversity of 69 species. Some species are presented within the Appendix A. Of these species, 94% are endemic to Madagascar, 10% are locally endemic, and 21% are regionally endemic. From this overall result, in 2014, we identified 10 amphibians of three families and 35 reptiles of six families. In 2020, we found 17 amphibians from two families and 44 reptiles from six families (Table 1). In 2014, we only investigated the Andavakoera site. However, in 2020, we recorded amphibians and reptiles from Sahabedara (14 amphibians/37 reptiles), Ampitiliantsambo (15 amphibians/30 reptiles), and Andavakoera (8 amphibians/24 reptiles) (Figure 2). We noticed that Andavakoera had fewer species recorded in 2020 than in 2014.

### 3.2. Abundance and Ecological Preference 

We found that 9 species (13%) were abundant, 21 species (30%) were common, 17 species (24%) were infrequent, and 22 species (31%) were rare (Table 1). Of the recorded amphibians, *Mantella viridis* was dominant (34% of the total) in Andavakoera, whereas other sites were dominated by *Boophis tephraeomystax*, *Mantidactylus bellyi*, and *Mantidactylus ulcerosus*. We also observed that 28 species were strictly arboreal (37%), 32 were in terrestrial environments (43%), 10 were semi-aquatic (13%), and 5 were found in rocky environments (7%). Half of the observed species (50%) were found within forested areas, 10% were recorded in anthropogenically disturbed areas, and 40% were in both forested and anthropogenically disturbed areas.

### 3.3. Newly Recorded Species at Montagne des Français

We identified three frogs (*Boophis* cf. *occidentalis*, *B.* cf. *marojezensis*, and *B.* sp.) and one snake (*Phisalixella* cf. *arctifasciata*) previously unknown to the area. Additionally, we found *Langaha pseudoalluaudi*, which previously had not been seen in the area since 2007. Below, we present a description of these observations:

#### 3.3.1. *Boophis* cf. *occidentalis*

This is a medium-sized deciduous forest species (SVL = 44.3 mm). The seven individuals observed had a green dorsum and yellow lateral line running between the snout tip and inguinal region. All the ventral surfaces were pink (Figure 2). The webbing formula is 1 (0.25) 2i (0.5) 2e (0.25) 3i (0.25) 3e (0.25) 4i (1) 4e (1) 5 (0). Males have evident tubercles, especially on the dorsum, which are absent in females. We observed this species on a branch overhanging the stream and above the riverbank from 6–9 p.m.

#### 3.3.2. *Boophis* cf. *marojezensis*

This is a small forest species (SVL = 23.5 mm) found close to a small stream. One individual had a cream color on the dorsum and a blue peripheral ring on its iris. The fingertips and toe tips were pigmented with a golden color. The webbing formula is 1 (0.5) 2i (0) 2e (0.25) 3i (1) 3e (0.5) 4i (1) 4e (1) 5 (0) and this characteristic shows some differentiation according to the webbing formula by Glaw et Vences (2007). This species was resting on a leaf of a shrub overhanging a small stream at 9:15 p.m. 

#### 3.3.3. *Boophis* sp.

This is a large forest species (SVL = 65.5 mm). This species could belong to the *Boophis goudotii* group based on the presence of a turquoise-blue outer iris periphery and a large SVL. The webbing formula is 1 (0.25) 2i (0) 2e (0) 3i (0.5) 3e (0.25) 4i (0.75) 4e (0.75) 5 (0). It was resting on a branch of a shrub overhanging a small stream at 10:15 p.m. 

#### 3.3.4. *Phisalixella* cf. *arctifasciata*

This arboreal snake was found at Montagne des Français during our 2014 and 2020 investigations. It is a large species (TL: 864 mm; SVL: 288 mm) but differs from *P. arctifasciatus* by the presence of 135 dark transverse bands between the neck and the tail tip. It was active during the night, around 11:00 p.m., on a branch at a height of 3.5 m next to a small stream. 

#### 3.3.5. *Langaha pseudoalluaudi*

This is a large arboreal forest species (TL = 1275 mm for males; 1260 m for females). These male and female individuals were found in the same environment. Both specimens were discovered under identical conditions, on shrubby plants measuring 2 m and 2.5 m at the periphery of the relict forest, situated in an open environment at an altitude of 80 m. 

## 4. Discussion

Our study provides new data on the distribution and endemism of amphibians and reptiles within Montagne des Français PA. The results build on previous studies [7,8,9,36] of the unique herpetofaunal community of Montagne des Français. All these studies have already highlighted the level of endemicity and habitat loss due to various threats, including the proximity of this zone to human settlements and their activities. 

### 4.1. Species Richness 

Incredible faunal diversity was found in the extreme northern karst formations of Madagascar. Sixty-nine species were encountered during this study. Of these, we recorded 10 amphibians from three families and 35 reptiles from six families in 2014 in Andavakoera. In addition, the survey we conducted in 2020 allowed us to encounter 16 amphibians from two families and 44 reptiles from six families. The endemicity level was 94%, which is similar to that in [7,9,34]. Despite the relatively short fieldwork, we were able to increase sampling effort by having a large research team composed of eight people. Moreover, we investigated other sites, such as Sahabedara and Ampitiliantsambo, located in the peripheral areas of Montagne des Français PA, in addition to Andavakoera, to further assess the herpetofaunal diversity of the area. Although these additional sites were not frequently surveyed, we found a higher species diversity of amphibians and reptiles than that of Andavakoera during the survey of 2020 (Table 1), including the highly cryptic snake *Langaha pseudoalluaudi* rediscovered in Sahabedara, which was last recorded by [7] in 2007. Furthermore, most of the newly recorded species were discovered in these sites, namely, *Boophis* cf. *marojezensis*, *B.* cf. *occidentalis*, *B.* sp., and the snake *Physalixella* cf. *arctifasciata*, which morphologically present a big challenge for classification and Montagne des Francais appears as a new biogeographical distribution for them. Importantly, we emphasize the need for bioacoustic and molecular studies on the newly recorded species we present here, as we cannot confirm their status based on morphology and appearance alone.

Comparing previous research data with those gathered during the current inventory, it can be shown that *Zonosaurus aenus*, *Thamnosophis lateralis*, and *T. stumpffi* were not identified by [7,9,25]. Relatedly, the recent study conducted by [25] did not confirm the occurrence of *Dromicodryas bernieri* and *Langaha pseudoalluaudi*, while the previous studies in 2007 [7,9] did. The absence of these animals could be attributed to seasonal factors. If surveys were conducted when they are inactive, it may make observing them challenging or impossible. Additionally, the lack of detection in some years might be due to survey differences between the western slope, Andavakoera, and the eastern slope, Sahabedara and Ampitiliantsambo, or due to the presence of bushfire, charcoal production, or other habitat factors affecting species presence during surveys.

### 4.2. Importance of Conservation

While the Montagne des Français is located near the urban area of Antsiranana, anthropogenic activities such as charcoal production [7] and timber harvesting provoke threats to species habitat and negatively affect herpetofauna [36]. Moreover, the sites of Sahabedara and Ampitiliantsambo were more disturbed by land use leading to a fragmented and degraded forest. Some species of herpetofauna in tropical dry forests have been shown to be highly resilient [37]. Indeed, certain species are found in forests of varying quality. For example, *Langaha pseudoalluaudi*, *Lycodryas* sp., *Sanzinia volontany*, and *Madagascarophis colubrinus* may be able to adapt their behavior to disturbed habitats. We observed these species frequently in areas with closed canopies and in more open habitats. At the same time, other species are more sensitive to minor habitat changes; for example, *Mantella viridis*, *Boophis* cf *majori*, *B.* cf *occidentalis*, and *B.* sp., which are typically restricted to intact humid forest. We suggest that the frequent and continued exploitation of forest will drastically reduce the habitat of amphibians and reptiles, meaning that they will be more prone to extinction [7,13,15]. The most threatened species in Montagne des Français are three amphibian species, *Mantella viridis* (EN), *Stumpffia roseifemoralis* (EN), and *Stumpffia staffordi* (VU), and nineteen reptile species, including three critically endangered, five endangered, and eleven vulnerable (cf. Table 1). To ensure the viability of the site, permanent monitoring and conservation planning is needed for all of them [21,38,39,40]. Particularly, we recommend the conservation of species that have a restricted distribution throughout the northern half of Madagascar, namely, *Mantella viridis*, *Paroedura lohatsara*, *Zonosaurus tsingy*, *Thamnosophis martae*, and *Heteroliodon fohy*.

## 5. Conclusions

The fieldwork conducted in 2014 and 2020 contributes to the existing literature regarding the diversity and occurrence of the herpetofaunal community of Montagne des Français, especially in Ampitiliantsambo, where investigation has not been undertaken before. Overall, this study highlighted the occurrence and the update of the distribution of 19 amphibian and 50 reptile species living within the Montagne des Français PA. In the face of a rapidly changing landscape, it is pivotal to consider an effective management and conservation plan for Montagne des Français regarding its proximity to Antsiranana city and frequent land use. Human activity in the boundary of this zone is one of the factors that drives habitat loss and consequently species decline. Intensive long-term monitoring has to be planned to assess herpetofaunal trends in the northern area of Madagascar, informing effective policies.

## Figures and Tables

**Figure 2 animals-13-03361-f002:**
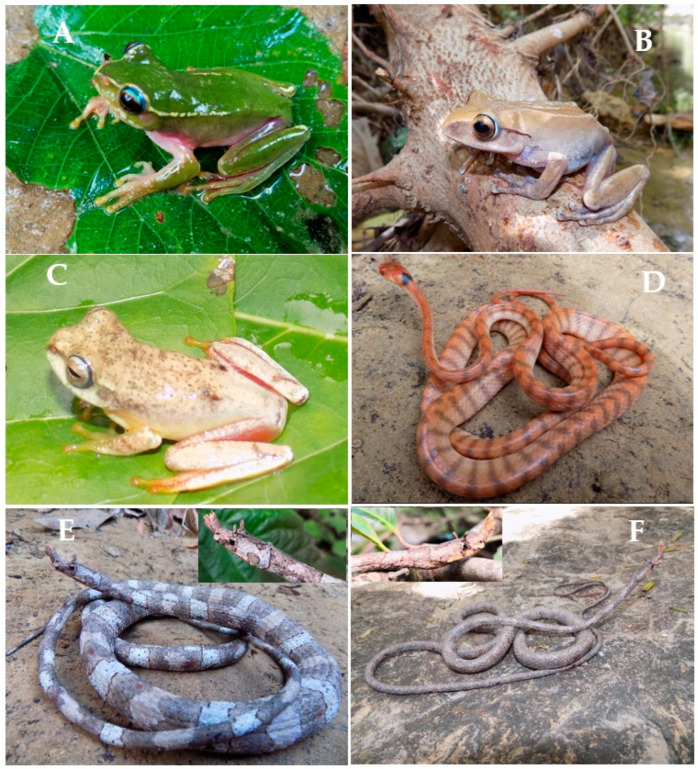
Photographic documentation of the morphospecies found and rediscovered species in the Montagne des Français area. (**A**): *Boophis* cf. *occidentalis*; (**B**): *Boophis* sp.; (**C**): *Boophis* cf. *marojezensis*; (**D**): *Phisalixella* cf. *arctifasciata*; (**E**): *Langaha pseudoalluaudi* (male); (**F**): *Langaha pseudoalluaudi* (female).

**Table 1 animals-13-03361-t001:** Species composition of the herpetofaunal community recorded in the NAP Montagne des Français during the years 2014 and 2020. Site: AND: Andavakoera, AMP: Ampitiliantsambo, SAH: Sahabedara. End: endemicity, IUCN: CR = Critically Endangered, EN = Endangered, VU = vulnerable, LC = Least Concern, NE = Not Evaluated. Relative abundance (RA): A = abundant, C = common, I = infrequent, R = rare. Ecological distribution (ED): Ab = arboreal, Tr = terrestrial, S = semiaquatic, Ro = rocky. Habitat: F = forest relatively intact, A= anthropogenically disturbed area. Endemicity (END): E = endemic, RE = regionally endemic, LE = locally endemic, NE = not endemic.

	2014	2020			IUCN	RA	ED	Habitat	END
TAXON	AND	AND	AMP	SAH					
AMPHIBIANS									
MANTELLIDAE									
*Aglyptodactylus securifer*	+	−	−	−	LC	I	Tr	F	E
*Boophis marojezensis*	−	−	+	−	NE	R	Ab	F	E
*Boophis occidentalis*	−	−	+	+	NE	R	Ab	F, A	E
*Boophis* sp.	−	−	−	+	NE	R	Ab	F, A	E
*Boophis tephraeomystax*	+	+	+	+	LC	A	Ab, S	F, A	E
*Gephyromantis pseudoasper*	−	−	+	+	LC	R	S	F	E
*Laliostoma labrosum*	+	−	+	−	LC	I	Tr	F, A	RE
*Mantella viridis*	+	+	+	+	EN	A	Tr	F, A	LE
*Mantidactylus bellyi*	+	+	+	+	LC	A	Tr	F, A	E
*Mantidactylus betsileanus*	+	−	−	−	LC	I	Tr	F	E
*Mantidactylus ulcerosus*	+	+	+	+	LC	A	Tr	F, A	E
PTYCHADENIDAE									
*Ptychadena mascareniensis*	+	+	+	+	LC	C	Tr, S	F, A	C
MICROHYLIDAE									
*Stumpffia analamaina*	−	−	+	+	LC	I	S	F	E
*Stumpffia angeluci*		+	+	+	LC	C	S	F	RE
*Stumpffia gimmeli*	+	−	−	+	LC	C	S	F	RE
*Stumpffia* cf. *madagascariensis*	−	−	+	+	LC	R	S	F	RE
*Stumpffia roseifemoralis*	−	+	+	+	EN	C	S	F	E
*Stumpffia staffordi*	−	+	+	+	VU	R	S	F	E
DICROGLOSSIDAE									
*Hoplobatrachus tigerinus*	+	−	−	−	LC		Tr	F, A	NE
REPTILES									
BOIDAE									
*Acrantophis madagascariensis*	−	−	+	+	LC	R	Tr	F, A	E
*Sanzinia volontany*	+	−	−	+	LC	C	Ab	F	E
CHAMAELEONIDAE									
*Brookesia ebenaui*	+	−	−	+	VU	I	Tr	F	E
*Brookesia stumpffi*	+	+	−	−	VU	C	Tr	F	E
*Brookesia tristis*	+	−	−	−	NE	R	Tr	F	E
*Furcifer oustaleti*	+	−	+	−	LC	A	Ab	F, A	E
*Furcifer pardalis*	+	−	+	−	LC	C	Ab	F, A	RE
*Furcifer petteri*	+	+	+	+	VU	I	Ab	F, A	E
GEKKONIDAE									
*Blaesodactylus boivini*	+	+	+	+	VU	C	Ab	F, A	RE
*Geckolepis maculata*	+	−	+	−	NE	C	Ab	F, A	E
*Geckolepis typica*	+	+	−	−	LC	C	Ab	F, A	E
*Hemidactylus frenatus*	+	+	−	+	LC	A (	Ab, Ro	A	NE
*Hemidactylus mercatorius*	+	−	−	−	LC	A	Ab, Ro	A	NE
*Lygodactylus heterurus*	−	−	+	+	LC	R	Ab	F	RE
*Paroedura hordiesi*	+	−	−	+	CR	R	Ro	F, A	LE
*Paroedura lohatsara*	+	+	+	+	CR	R	Ro	F, A	LE
*Paroedura stumpffi*	+	−	−	+	LC	C	Ro	F, A	RE
*Phelsuma abbotti*	+	+	+	+	LC	I	Ab	F	E
*Phelsuma grandis*	+	−	+	−	LC	A	Ab	F	RE
*Uroplatus fetsy*	−	−	+	+	NE	R	Ab	F	RE
*Uroplatus giganteus*	−	−	−	+	VU	R	Ab	F	E
*Uroplatus henkeli*	+	−	+	−	VU	C	Ab	F	E
*Uroplatus sameiti*	+	+	−	−	LC	C	Ab	F	E
LAMPROPHIDAE									
*Alluaudina bellyi*	−	+	−	−	LC	R	Tr	F	E
*Dromicodryas bernieri*	−	−	+	+	LC	C	Tr	A	E
*Dromicodryas quadrilineatus*	+	−	+	+	LC	C	Tr	A	E
*Heteroliodon fohy*	+	−	−	−	EN	R	Tr	F	LE
*Ithycyphus miniatus*	+	−	+	+	LC	I	Ab	F	E
*Langaha madagascariensis*	+	−	−	−	LC	I	Ab	F, A	E
*Langaha pseudoalluaudi*	−	−	−	+	LC	R	Ab	F, A	E
*Leioheterodon modestus*	+	−	−	+	LC	C	Tr	F, A	E
*Leioheterodon madagascariensis*	+	+	+	+	LC	C	Tr	F, A	E
*Liophidium therezieni*	+	−	−	−	EN	R	Tr	F	LE
*Liophidium torquatum*	+	−	+	+	LC	I	Tr	F	E
*Lycodryas inopinae*	+	−	+	+	LC	R	Ab	F, A	RE
*Lycodryas granuliceps*	−	−	+	−	LC	I	Ab	F, A	E
*Lycodryas pseudogranuliceps*	−	−	−	+	LC	C	Ab	F, A	E
*Madagascarophis colubrinus*	+	+	+	+	LC	C	Tr	F, A	E
*Madagascarophis fuchsi*	+	+	−	−	CR	I	Tr	F	LE
*Mimophis occultus*	−	+	+	+	LC	C	Tr	F, A	E
*Phisalixella* cf. *arctifasciata*	+	−	+	−	NE	R	Ab	F	LE
*Phisalixella arctifasciata*	+	−	+	+	LC	I	Ab	F	RE
*Pseudoxyrhopus quinquelineatus*	+	−	−	+	LC	I	Tr	F	RE
*Thamnosophis martae*	−	+	+	−	EN	I	Tr	F	RE
*Thamnosophis stumpffi*	−	−	+	−	VU	I	Tr	F	E
SCINCIDAE									
*Madascincus miafina*	−	−	−	+	LC	R	Tr	F	E
*Trachylepis elegans*	+	+	+	+	LC	A	Tr	A	E
*Trachylepis tavaratra*	−	−	−	+	VU	C	Tr	A	RE
*Voeltzkowia* sp.	+	−	−	−		R	Tr	A	
TYPHLOPIDAE									
*Indotyphlops braminus*	+	−	+	+	NE	R	Tr	F	E
Total of Amphibians	10	8	14	14					
Total of Reptiles	36	24	30	37					

## Data Availability

The data presented in this study are available on request from the corresponding author. The data are not publicly available due to privacy or ethical restrictions.

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
