# Peer review of "Amphibians and Reptiles of the Montagne des Français: An Update of the Distribution and Regional Endemicity"

_animals, 2023, doi:10.3390/ani13213361_

Round 1

Reviewer 1 Report

Comments and Suggestions for Authors

Comments on the Quality of English Language

As a native speaker, CJR needs to read and correct the English.

Reviewer 2 Report

Comments and Suggestions for Authors

I have a few comments about the manuscript:

- The abstract does not fully describe the research results and needs to be corrected

 - The introduction is too short. You need to add additional publications to it

- How do you explain such a small amount of time that you spent doing research? In such a short period, is it possible to study the entire territory of the protected area? Why didn't acoustic research methods be used? Why were molecular and genetic analyses not used?

- Figure 2 and Table 1 to some extent repeat each other. Redo the drawing and the table so that they do not repeat.

- You describe in the manuscript potentially new species for science. However, you should understand that potentially new species does not mean that they are definitely new species. Therefore, it makes no sense to describe these species in the manuscript in this form. You have to describe new species and it will be more effective for the manuscript than the assumptions about novelty put forward. At the same time, you must describe them according to all the rules of the Code of International Nomenclature.

- Indications of endemism and biogeographic distribution in the discussion are not confirmed in the results. Also, the methods do not specify the publication on the basis of which the biogeographic distribution of amphibians and reptiles is determined.

- The discussion is written on the principle of observation and many of the provisions in the discussion are not confirmed by other publications. Thus, many descriptions are unfounded. 

- In general, in the discussion, the authors describe facts that they do not indicate in any way in the results. For example, anthropogenic impact or endemism.

- In conclusion, the authors provide little information to understand the research results.

Reviewer 3 Report

Comments and Suggestions for Authors

I reviewed your manuscript “Amphibians and Reptiles of the “Montagne des Français”: update of the distribution and regional endemicity” showing results from herpetological inventories in 2014 and 2020. Importantly, you observed several species previously unreported from the area and documented the presence of threatened taxa. With some work, the manuscript can be revised for publication, but in its current form it is not ready.

Here are the main areas to work on:

1)      New species identification:  Currently, I cannot tell if section three of the results is documenting new species in the reserve or trying to describe new species to science. Certainly, the evidence presented in your manuscript, such as morphometrics, is not enough to describe new species. So, I think the aim is to say there are these undescribed species previously unrecorded at Montagne des Français. I suggest acknowledging the possibility of these species being undescribed or new and emphasizing the need for further research, including molecular and bioacoustics data, to confirm their status. To avoid any confusion, consider editing or removing figures 3–7 that may imply the observation of undescribed species, since really you don’t know what they are. Focus on reporting the species that are new to the reserve, not stating that they are new to science (because you don’t know if they are or not).

2)      Detailed methods: The methods lack enough detail for somebody else to reproduce your surveys. For example, how long were your drift fences? How many pitfalls did each have? How often did you check them? What is this survey method you mention about microhabitat refugia? How many surveys did you perform of each kind at each site? How long did surveys last? I suggest having a paragraph about each survey method. Then, include enough information so that somebody else in 10, 20, 30 years could repeat your survey methods at the site.

3)      Briefly, review previous literature in the intro. What have prior inventories found at Montagne des Francais? How many species did they find? What do we know already and what do we need to know that we don’t know yet? Build a knowledge gap.

4)      Interpretation of figures and tables: I have provided comments on the PDF regarding the interpretation of figures and tables. For instance, define abbreviations like "EL" in Table 1 and explain any colored lines or symbols in the figures, like the pink line in Figure 1. Change the fill of the bar graph so we can see they are different.

5)      Grammar and Technical Issues: The manuscript requires significant editing to address grammar and technical errors. The errors are substantial enough that they make it difficult to understand the study and results. Try running the text through Grammarly (https://www.grammarly.com/) or other grammar checker (maybe this one https://quillbot.com/grammar-check) to clean it up. Here are some common minor issues to watch out for as you continue writing:

a.       Articles like “the” and “a”, for example lines 19. See https://owl.purdue.edu/owl/general_writing/grammar/using_articles.html

b.      Plural/singular forms, for example “was” versus “were” line 20, line 25, etc.

c.       Proper use of verb tenses, avoiding the progressive tense for actions that are not continuing. See https://owl.purdue.edu/owl/general_writing/mechanics/gerunds_participles_and_infinitives/participles.html

d.       Capitalize only proper nouns, for example “Site manager” should be “site manager” line 22, “Herpetofauna” should be “herpetofauna” line 28, “Reptile” should be “reptile”, etc. See https://owl.purdue.edu/owl/general_writing/mechanics/help_with_capitals.html

e.       Numbers: write the number in letters if it is below 10 and not a measured quantity, for example five sites. Use numbers if measured or 10 and above, 13 species. Also, do not put a 0 before numbers less than 10.

f.        Define acronyms before use. For example, E, ER, EL in Table 1, NAP, etc.

g.       There are errors in the references, for example the article title of 6 is all uppercase

Please also see my 17 comments directly on the PDF that I uploaded.

Additionally, I recommend reviewing other publications reporting inventory results. Then, structure your article in a similar way. For example, here are three articles that present inventory results without molecular data:

https://www.herpconbio.org/Volume_6/Issue_1/Durkin_etal_2011.pdf https://www.herpconbio.org/Volume_4/Issue_1/D'Cruze_etal_2009b.pdf https://www.herpconbio.org/Volume_6/Issue_2/Rakotondravony_Goodman_2011.pdf

You can use these three articles as a guide while revising your manuscript. Make your writing more like the writing in the above three papers. Especially, please look at the detail of the methods section and the type of results they report. They find possibly undescribed species, but they do not focus on them. Instead, focus on species richness and how your results compare to previously published inventories.

With significant revisions and thorough editing, your manuscript has the potential to become a high-quality publication. I hope my feedback is helpful as you continue making edits.

Comments on the Quality of English Language

Please see my above comments. Many parts of the manuscript are difficult to understand. The organization is okay, but the things that make it hard to understand relate to sentence structure, grammar, and technical issues. See if Grammarly helps.

Round 2

Reviewer 2 Report

Comments and Suggestions for Authors

Dear authors. Thank you for the answers. You have made ruokpisi adjustments. It is very good. However, there remains the question that I pointed out. You refer that you have found new species for science. But at the same time, there is no primary description of these species in the submitted manuscript. If you don't describe these species, then you don't need to write about them at all. Why do this? You want to improve your manuscript, give it "weight". But then describe these species with all the rules of international nomenclature. Or don't mention such species at all until you describe them. In general, you need to make adjustments to the text again.

Author Response

Dear reviewer

Best regards,

Nirhy RABIBISOA

Reviewer 3 Report

Comments and Suggestions for Authors

I recommend major revisions, all of which are related to the writing. In the attached PDF, I have provided 83 comments on the writing itself to improve readability.

Comments on the Quality of English Language

I recommend major revisions, all of which are related to the writing. In the attached PDF, I have provided 83 comments on the writing itself to improve readability.

Author Response

Dear reviewer

We are grateful to you and very satisfied about your rigorous reviews on our manuscript and thank you very much for taking the time to do this. Please find the detailed responses below and the corresponding revisions/corrections highlighted/in track changes in the re-submitted files

Best regards,

Nirhy RABIBISOA
